# Anti-HIV Potential of Beesioside I Derivatives as Maturation Inhibitors: Synthesis, 3D-QSAR, Molecular Docking and Molecular Dynamics Simulations

**DOI:** 10.3390/ijms24021430

**Published:** 2023-01-11

**Authors:** Zixuan Zhao, Yinghong Ma, Xiangyuan Li, Susan L. Morris-Natschke, Zhaocui Sun, Zhonghao Sun, Guoxu Ma, Zhengqi Dong, Xiaohong Zhao, Meihua Yang, Xudong Xu, Kuohsiung Lee, Haifeng Wu, Chinho Chen

**Affiliations:** 1Beijing Key Laboratory of New Drug Discovery Based on Classic Chinese Medicine Prescription, Key Laboratory of Bioactive Substances and Resources Utilization of Chinese Herbal Medicine, Ministry of Education, Institute of Medicinal Plant Development, Chinese Academy of Medical Sciences and Peking Union Medical College, Beijing 100193, China; 2Natural Products Research Laboratories, UNC Eshelman School of Pharmacy, University of North Carolina, Chapel Hill, NC 27599, USA; 3Antiviral Drug Discovery Laboratory, Surgical Oncology Research Facility, Duke University Medical Center, Durham, NC 27710, USA

**Keywords:** anti-HIV, maturation inhibitor, beesioside I, CA-SP1, 3D-QSAR, molecular docking, molecular dynamics simulations

## Abstract

HIV-1 maturation is the final step in the retroviral lifecycle that is regulated by the proteolytic cleavage of the Gag precursor protein. As a first-in-class HIV-1 maturation inhibitor (MI), bevirimat blocks virion maturation by disrupting capsid-spacer peptide 1 (CA-SP1) cleavage, which acts as the target of MIs. Previous alterations of beesioside I (**1**) produced (20*S*,24*S*)-15*ꞵ*,16*ꞵ*-diacetoxy-18,24; 20,24-diepoxy-9,19-cyclolanostane-3*ꞵ*,25-diol 3-*O*-3′,3′-dimethylsuccinate (**3**, DSC), showing similar anti-HIV potency compared to bevirimat. To ascertain the binding modes of this derivative, further modification of compound **1** was conducted. Three-dimensional quantitative structure–activity relationship (3D-QSAR) analysis combined with docking simulations and molecular dynamics (MD) were conducted. Five new derivatives were synthesized, among which compound **3b** showed significant activity against HIV-1_NL4-3_ with an EC_50_ value of 0.28 µM. The developed 3D-QSAR model resulted in great predictive ability with training set (*r*^2^ = 0.99, *q*^2^ = 0.55). Molecular docking studies were complementary to the 3D-QSAR analysis, showing that DSC was differently bound to CA-SP1 with higher affinity than that of bevirimat. MD studies revealed that the complex of the ligand and the protein was stable, with root mean square deviation (RMSD) values <2.5 Å. The above results provided valuable insights into the potential of DSC as a prototype to develop new antiviral agents.

## 1. Introduction

Despite the great success of highly active antiretroviral therapy (HAART) at reducing mortality from HIV infection, currently available antiretroviral medications frequently fail to act as an eradicative cure, which are usually accompanied by different degrees of toxic side effects and drug resistance. Therefore, it is urgent to develop anti-HIV drugs with novel mechanisms of action. Bevirimat (BVM), derived from betulinic acid, is a first-in-class HIV-1 maturation inhibitor (MI), which exhibits potent effects against viruses resistant to nucleoside and non-nucleoside reverse transcriptase inhibitors (NRTIs and NNRTIs), protease inhibitors (PIs), and integrase inhibitors (INs), and even displays synergistic effects with other anti-HIV drugs [1,2]. Because bevirimat possesses a favorable safety profile and mode of action distinct from other antiretroviral agents as well as prominent antiretroviral activity, BVM and its analogues were subjected to extensive studies and entered phase IIb clinical trials, successively [3,4]. In this situation, natural products and their structural analogues could be seen as a necessary complement to conventional HAART and thus have attracted much more attention [5,6].

HIV-1 maturation is the final step in the viral replication cycle that is governed by the proteolytic cleavage of the Gag precursor protein [7]. The Gag polyprotein is stepwise split by viral proteases into matrix (MA), capsid (CA), spacer peptide 1 (SP1), nucleocapsid (NC), spacer peptide 2 (SP2) and p6, which subsequently rearrange to form mature and infectious virion. The cleavage between CA and SP1 is a crucial rate-limiting step, which acts as the target of HIV-1 MIs [8]. The structural studies of CA-SP1 have shown that the C-terminal domain (CTD) of CA and SP1 forms a six-helix bundle (6HB), which holds together the Gag hexamer necessary for HIV particle assembly [9,10,11]. The 6HB actually works as molecular switch to control the maturation process [12]. Bevirimat works by binding to the CA–SP1 peptide and stabilizing 6HB and thus keeping the enzyme from cutting the Gag polyprotein [13]. The accumulation of CA–SP1 intermediate with the absence of assembled CA proteins led to loss of infectivity [14]. Further investigation revealed that BVM binds at the center of 6HB with the axial orientation via electrostatic and hydrophobic interactions [15]. Figure 1 depicts the structure of the HIV-1 Gag precursor protein and BVM binding to CA_CTD_-SP1.

In silico molecular modeling has played an increasingly pivotal role in contemporary drug discovery [16]. Computer experiments using various computational techniques and software programs have proven to be essential for the interpretation of common experimental assays [17]. Nowadays, 3D-quantitative structure–activity relationship (3D-QSAR) models have been intensely applied in drug design [18]. Virtual screening and/or molecular docking software program as well as molecular dynamics (MD) have been extensively used to guide the drug discovery [19,20]. In particular, MD simulations have become increasingly useful for understanding the structure–function relationship of the target and the essence of protein–ligand interactions [21,22]. With the advent of high-performance computing, graphics processing units (GPUs)-based MD simulations are capable of unveiling biological phenomena that were previously not feasible using traditional hardware architectures [23]. A growing trend in the application of 3D-QSAR combined with molecular docking and MD simulation study in drug design has been witnessed in recent years [24,25].

Triterpenoids and their analogs have been extensively investigated for their diverse structures and antiviral activities [26,27,28]. In our previous efforts to search for new anti-HIV agents from natural products, beesioside I (**1**) was found to show significant antiviral effects. The following modification on aglycone (**2**) of compound **1** led to the discovery of (20*S*,24*S*)-15*ꞵ*,16*ꞵ*-diacetoxy-18,24; 20,24-diepoxy-9,19-cyclolanostane-3*ꞵ*,25-diol 3-*O*-3′,3′-dimethylsuccinate (**3**, DSC) with EC_50_ value of 0.025 μM, displaying similar anti-HIV potency compared to bevirimat [29]. However, the plausible binding mode of the promising derivative to CA-SP1 remains unknown. In this study, compound **1** was further modified to produce five new derivatives **2a**–**b** and **3a**–**c** among which compound **3b** showed the strongest activity against HIV-1_NL4-3_ with an EC_50_ value of 0.28 µM (CC_50_ > 20 µM). To further explore the potential of beesioside I derivatives as anti-HIV MIs, 3D-QSAR modeling was developed using the dataset consisting of 14 anti-HIV cycloartane triterpenoids and analogues, including DSC. Molecular docking techniques were utilized to further explain the results of the 3D-QSAR study and explore the possible binding modes. In addition, to further prove the reliability of the docking results and confirm the stability of the complex of ligand and protein, a post-docking MD simulations study was also performed. We believe that the results presented in this study can provide in-depth information for the development of new antiviral agents from triterpenes.

## 2. Results and Discussion

### 2.1. Chemistry

Figure 1 depicts the syntheses of the derivatives from **1** (**2a**–**b** and **3a**–**c**). Acylation of the 3-OH of compound **2** with benzoyl chloride and croton anhydride produced compounds **2a**–**b**. Deacetylation and benzylation of compound **3** gave **3a**–**b**. Furthermore, deacetylation of **3b** with potassium hydroxide in ethanol yielded **3c**.

### 2.2. Evaluation of Anti-HIV Activity

All of the synthetic derivatives were evaluated for anti-HIV activity by determining the inhibitory effects on virus replication in MT4 lymphocytes infected by HIV-1_NL4-3_ Nanoluc-sec virus. Cytotoxic activity tests were also carried out on MT4 cells. Table 1 shows the anti-HIV and cytotoxic results for the tested compounds with AZT as the positive control. Compound **3b** showed the best potency among the derivatives.

### 2.3. 3D-QSAR Study

To obtain in-depth understanding of the positive and negative effects of modifications at different sites on antiviral activity, a position-based 3D-QSAR analysis was conducted using the Schrödinger Suites 2018. The 3D-QSAR models were constructed and validated using the dataset composed of 14 cycloartane triterpenoids and analogues with their corresponding activity data reported from our laboratory. As shown in Table 2, 14 small molecules were randomly divided into a training set with 11 molecules and a test set with 3 molecules. The assembled 3D-QSAR model has the following Gaussian field fractions: 0.28 (steric), 0.11 (electrostatic), 0.20 (hydrophobic), 0.22 (hydrogen bond acceptor) and 0.19 (hydrogen bond acceptor). It exhibits a good correlation coefficient (*r*^2^) of 0.99 and reliable predictivity with a cross-validation correlation coefficient (*q*^2^) of 0.55. The linear correlation between the experimental results and the predicted values are shown in Figure 2 and listed in Table 2.

The QSAR result was further visualized by 3D contour maps to determine the influence of steric and electrostatic fields on the activity value of the compounds and DSC was used as the template molecule in the map. The model of the steric field is shown in Figure 3a, where the favorable region of the larger steric field (in green) is located near the carboxyl group of the side chain and the D-ring linked acetyl group, suggesting that the introduction of large groups in this region could increase the activity. The yellow part suggests that the introduction of small groups could improve the activity. Figure 3b is a model of the electrostatic field, with positive coefficients shown in blue and negative coefficients in red. The dimethyl succinyl group of the C-3 side chain and the D-ring linked acetyl group as well as the hydroxyl group at position 25 are covered in red, implying that the negative charge in this region is favorable for increased activity. The blue outline shows that increasing the positive charge in this region will affect the activity positively.

### 2.4. Molecular Docking Study

The molecular docking studies were conducted to elaborate the 3D-QSAR results and clarify the possible binding modes of target compounds with the active site of CA_CTD_-SP1. DSC was selected as the ligand example and BVM as the positive control. DSC showed a better docking score of –8.096 kcal/mol (RMSD 2.73 Å) as compared with the standard BVM (docking score = –6.574 kcal/mol, RMSD = 1.56 Å), suggesting that DSC has a higher affinity for the CA-SP1 peptide than BVM. The docking diagram of BVM/DSC with CA-SP1 was analyzed by the overall docking graph and three-dimensional interaction map in detail. The results showed that BVM was vertically bound to the center of the 6HB in a linear extended conformation with the C-3 side chain oriented toward the upper part of the goblet-shaped hexamer as presented in Figure 4a,c, consistent with the literature data [30]. The binding of BVM and CA-SP1 was mainly mediated through electrostatic interactions and hydrophobic interactions. The terminal carboxylic group of the C-3 side chain of BVM formed two ionic bonds with two amino acids (LYS290 and LYS359). BVM also came into contact with LEU363 and MET367 via hydrophobic interactions, as shown in Figure 5a,b. In contrast, DSC was slightly inclined to the center of the 6HB as depicted in Figure 4b,d. In the binding mode (Figure 5c,d), DSC was bound to CA-SP1 via two ionic bonds and hydrophobic interactions. Moreover, the acetyl group at 16-position of DSC formed a hydrogen bond with LYS359. The above results suggested that DSC could interact in a different way with the CA_CTD_-SP1 junction helix compared to BVM.

### 2.5. Molecular Dynamics

MD studies were carried out to integrate the results of the docking simulation with the analysis of molecular movements of CA-SP1 upon ligands binding. The docking poses of DSC and BVM with CA-SP1 underwent 50 ns. The kinetic energy and potential energy tended to be stable during the MDs process, as depicted in Appendix A. The total energy change in the simulation process is represented by a frequency distribution diagram. It can be observed that the energy distributions of both DSC and BVM are normal (Appendix A). In addition, the Awk script is used to search for the frame number corresponding to the lowest potential energy in the MDs process. The lowest potential energy structures are shown in Appendix A. Figure 6 shows that the RMSD of the reference BVM fluctuated between 0.75 and 1.75 Å, accompanied by the RMSD of DSC fluctuated between 1.00 and 2.25 Å, which was slightly larger than that of BVM. Overall, the RMSD value fluctuated less during the MD simulation, indicating that the docking poses were reliable. Visualization of the MD simulation for DSC was performed using the VMD analysis tool, seen in Figure 7.

In order to further analyze the free energy of DSC and CA-SP1, molecular mechanics/generalized born surface area (MM/GBSA) and molecular mechanics/Poisson–Boltzmann surface area (MM/PBSA) methods were used for the calculation of the total binding energy. The energy data of BVM and DSC were carried out as listed in Table 3. The absolute values of total binding free energy of the complex of DSC-CA-SP1 obtained by the two calculation methods (–48.27 and –48.18 kcal/mol) were accordingly higher than that of BVM (–38.13 and –42.79 kcal/mol), suggesting that DSC had stronger binding ability with CA-SP1 peptide. For both compounds, all of the VDW, EE, polar and apolar energy made necessary contributions for the binding.

With the purpose of exploring the key amino acid residues around ligands in CA-SP1 protein binding, the energetic decomposition of amino acid residues was performed. Figure 8 shows amino acid residues with energy contribution values lower than −2 kcal/mol, indicating that LYS359, LYS290 and LEU363 played a major role in binding free energy. Furthermore, the alanine binding site-scan (ABS-scan) was conducted to quantitatively characterize the binding free energy of specific residues in protein–ligand binding. The results showed that the strongest interaction was brought about by the residue LYS359 with a total fraction of −7.96 kcal/mol followed by LYS290 with a calculated value of −4.45 kcal/mol. The effect of LEU363 was mainly composed of van der Waals forces with a calculated value of −2.08 kcal/mol. All of these results are consistent with the molecular docking experiments. Because all the binding free energy changes caused by the three amino acid mutations are less than −2 kcal/mol, residues LYS359, LYS290 and LEU363 can be defined as hot-spot amino acids. The contributions of each energy component are shown in Figure 9.

## 3. Materials and Methods

### 3.1. Chemistry

All chemical reagents and solvents were obtained from Sigma Aldrich or other commercial source and used without further purification. ^1^H and ^13^C NMR spectra were measured in pyridine-*d*_5_ on Bruker AV III 600 NMR spectrometer (Bruker, Billerica, German) with TMS as internal standard. HRESIMS spectra were performed on an LTQ Obitrap XL spectrometer (Thermo Fisher Scientific, Boston, MA, USA). Melting points were determined on an X-4B apparatus (Shanghai Precision Instruments Co., Ltd., Shanghai, China) and are uncorrected. Optical rotations were obtained on an Anton Paar MCP200 polarimeter (Anton Paar, Graz, Austria). Semi-preparative HPLC on a CXTH LC3050N equipped with two pumps of P3000 and a UV3000 detector (Beijing ChuangXinTongHeng Science and Technology Co., Ltd., Beijing, China), and a ZORBAX Eclipse XDB-C18 (9.4 × 250 mm, 5 μm, Agilent, Santa Clara, CA, USA) column. Precoated silica gel GF_254_ plates (Zhi Fu Huang Wu Pilot Plant of Silica Gel Development, Yantai, China) and Merck precoated silica gel 60 F254 plates were used for TLC and spots visualized by heating plates sprayed with 10% H_2_SO_4_ in EtOH. Flash chromatograph was carried out on a Biotage Isolera Four system with a Buchi FlashPure Select Silica gel cartridge (15 μm, spherical, 4 g).

### 3.2. General Procedure for Synthesis of Compounds ***2a**–**b***

Compound **2** (0.017 mmol) in anhydrous pyridine (2.0 mL) was acylated separately with benzoyl chloride (0.02 mL) and crotonic anhydride (0.02 mL) with DMAP followed by chromatography to yield **2a** and **2b**, respectively. NMR and MS spectra of **2a**–**b** were provided in the Appendix A.

#### 3.2.1. (20*S*,24*S*)-15β,16β-Diacetoxy-18,24; 20,24-diepoxy-9,19-cyclolanostane-25-hydroxy-3I-yl benzoate (**2a**)

Colorless needles, 50.6% yield; m.p. 218–220 °C (MeOH); [α${]}_{D}^{20}$ 14.0 (*c* 0.1, MeOH); ^1^H NMR (600 MHz, pyridine-*d*_5_) *δ*_H_ 0.25 (1H, d, *J* = 4.0 Hz, H-19a), 0.54 (1H, d, *J* = 3.7 Hz, H-19b), 0.60 (1H, q, *J* = 11.7 Hz, H-6a), 0.94 (3H, s, H-28), 1.04 (3H, s, H-29), 1.05 (1H, m, H-7a), 1.16 (1H, m, H-1a), 1.16 (1H, m, H-11a), 1.22 (3H, s, H-30), 1.30 (3H, s, H-27), 1.30 (1H, m, H-7b),1.32 (1H, m, H-5), 1.34 (1H, m, H-6b), 1.54 (H, m, H-12a), 1.58 (1H, m, H-1b), 1.58 (3H, s, H-21), 1.65 (1H, dd, H-8), 1.70 (3H, s, H-26), 1.90 (1H, m, H-23a), 1.94 (1H, m, H-22a), 2.00 (1H, m, H-11b), 2.06 (1H, m, H-2a), 2.14 (3H, s, COCH_3_), 2.15 (1H, m, H-23b), 2.16 (3H, s, COCH_3_), 2.69 (1H, d, *J* = 11.5 Hz, H-17), 2.80 (1H, m, H-2b), 2.98 (1H, m, H-22b), 2.98 (H, m, H-12b), 4.54 (1H, d, *J* = 13.3 Hz, H-18a), 4.62 (1H, d, *J* = 13.2 Hz, H-18b), 5.02 (1H, dd, H-3), 5.72 (1H, d, *J* = 8.9 Hz, H-15), 5.98 (1H, dd, *J* = 11.5, 8.9 Hz, H-16), 7.52 (2H, t, *J* = 7.7 Hz, H-4′,6′), 7.60 (H, m, H-5′), 8.30 (2H, dd, *J* = 7.1, 1.3 Hz, H-3′,7′); ^13^C NMR (pyridine-*d_5_*, 150 MHz) *δ*_C_ 15.5 (C-29), 15.7 (C-30), 20.5 (C-9), 21.0 (2×COCH_3_), 21.5 (C-6), 25.8 (C-27,28), 26.0 (C-26), 27.1 (C-7), 28.4 (C-11), 28.6 (C-10), 29.2 (C-12), 30.9 (C-23), 31.3 (C-2), 32.1 (C-1), 32.6 (C-19), 33.4 (C-21), 39.5 (C-22), 41.2 (C-4), 46.7 (C-5), 47.0 (C-13), 48.0 (C-8), 52.7 (C-14), 56.3 (C-17), 67.6 (C-18), 74.0 (C-25), 75.2 (C-16), 81.2 (C-15), 82.0 (C-3), 88.7 (C-20), 115.6 (C-24), 129.0 (C-4′,6′), 130.0 (C-3′,7′), 132.7 (C-2′), 134.6 (C-5′), 167.5 (C-1′), 171.9 (COCH_3_), 172.3 (COCH_3_); HRESIMS *m/z* 715.3804 [M + Na]^+^ (Cacld. for C_41_H_56_O_9_Na *m/z* 715.3822).

#### 3.2.2. (20*S*,24*S*)-15β,16β-Diacetoxy-18,24; 20,24-diepoxy-9,19-cyclolanostane-25-hydroxy-3*β*-yl Crotonate (**2b**)

Colorless needles, 35.3% yield; m.p. 186–188 °C (MeOH); [α${]}_{D}^{20}$ 7.0 (*c* 0.1, MeOH); ^1^H NMR (600 MHz, pyridine-*d*_5_) δ_H_ 0.21 (1H, d, *J* = 4.1 Hz, H-19a), 0.52 (1H, d, *J* = 4.1 Hz, H-19b), 0.59 (1H, q, *J* = 12.4 Hz, H-6a), 0.94 (3H, s, H-28), 0.98 (3H, s, H-29), 1.05 (1H, m, H-7a), 1.12 (1H, m, H-1a), 1.12 (1H, m, H-11a), 1.21 (3H, s, H-30), 1.29 (3H, s, H-27), 1.31 (1H, m, H-7b), 1.25 (1H, m, H-5), 1.32 (1H, m, H-6b), 1.50 (H, m, H-12a), 1.49 (1H, m, H-1b), 1.58 (3H, s, H-21), 1.65 (1H, dd, H-8), 1.70 (3H, s, H-26), 1.65 (1H, m, H-2a), 1.74 (H, dd, H-3′), 2.08 (1H, m, H-23a), 1.97 (1H, m, H-22a), 1.96 (1H, m, H-11b), 1.97 (1H, m, H-2b), 2.82 (1H, m, H-23b), 2.14 (3H, s, COCH_3_), 2.16 (3H, s, COCH3), 2.75 (1H, d, *J* = 11.5 Hz, H-17), 2.99 (1H, m, H-22b), 2.96 (H, m, H-12b), 4.52 (1H, d, *J* = 13.2 Hz, H-18a), 4.60 (1H, d, *J* = 13.2 Hz, H-18b), 4.89 (1H, dd, H-3), 5.71 (1H, d, *J* = 8.9 Hz, H-15), 5.97 (1H, dd, *J* = 11.5, 8.9 Hz, H-16), 6.03 (1H, m, H-2′), 7.13 (1H, m, H-3′); ^13^C NMR (pyridine-*d*_5_, 150 MHz) δ_C_ 15.2 (C-29), 15.2 (C-30), 18.6 (C-9), 19.5 (C-6), 25.3 (C-27, 28), 25.6 (C-26), 25.8 (C-7), 26.8 (C-11), 27.0 (C-10), 27.6 (C-12), 29.9 (C-23), 30.8 (C-2), 31.2 (C-19), 31.5 (C-1), 32.2 (C-21), 38.0 (C-22), 39.5 (C-4), 45.5 (C-13), 46.5 (C-5), 46.7 (C-8), 52.2 (C-14), 55.9 (C-17), 66.1 (C-18), 72.5 (C-25), 74.8 (C-16), 79.7 (C-15), 81.7 (C-3), 86.6 (C-20), 114.1 (C-24), 129.0 (C-2′), 144.3 (C-3′), 165.9 (C-1′), 170.4 (COCH_3_), 170.8 (COCH_3_); HRESIMS *m/z* 657.3981 [M + H]^+^ (Cacld. for C_38_H_57_O_9_
*m/z* 657.4003).

### 3.3. General Procedure for Synthesis of Compounds ***3a**–**c***

Compound **3** (0.042 mmol) was treated separately with 2.5% KOH in EtOH (5 mL) and bromobenzene (0.01 mL) and potassium carbonate (12 mg) in DMF (5 mL) to give **3a** (36.8% yield) and **3b** (45.2% yield), respectively. Compound **3b** (0.012 mmol) was treated with 2.5% KOH in EtOH (3 mL) to give **3c** (53.5% yield). The solution was monitored by TLC until completed. The crude product was subjected to silica gel column chromatography followed by purification with semi-preparative HPLC using CH_3_CN/H_2_O as a mobile phase to give a colorless solid. NMR and MS spectra of **3a**–**c** were given in the Appendix A.

#### 3.3.1. (20*S*,24*S*)-18,24;20,24-Diepoxy-9,19-cyclolanostane-3*β*,15*β*,16*β*,25-tetraol-3-*O*-3′,3′-dimethylsuccinate (**3a**)

Colorless needles, 36.8% yield; m.p. 213–215 °C (MeOH); [α${]}_{D}^{20}$ 4.0 (*c* 0.1, MeOH); ^1^H NMR (600 MHz, pyridine-*d*_5_) δ_H_ 0.29 (1H, d, *J* = 4.0 Hz, H-19), 0.52 (1H, d, *J* = 4.0 Hz, H-19), 0.69 (1H, q, *J* = 12.0 Hz, H-6a), 0.95 (3H, s, H-28), 0.98 (3H, s, H-29), 1.04 (1H, m, H-7a), 1.15 (1H, m, H-1a), 1.25 (1H, m, H-7b, 11a), 1.26 (2H, m, H-12), 1.28 (1H, m, H-5), 1.31 (3H, s, H-30), 1.48 (3H, s, H-27), 1.49 (1H, m, H-1b, 6b), 1.55 (6H, s, 2×CH_3_-3′), 1.59 (3H, s, H-21), 1.64 (1H, m, H-2a), 1.66 (3H, s, H-26), 1.72 (1H, m, H-22a), 1.85 (1H, m, H-8), 1.92 (1H, m, H-2b), 2.02 (1H, d, *J* = 10.8 Hz, H-17), 2.04 (1H, m, H-11b), 2.05 (1H, m, H-23a), 2.06 (1H, m, H-22b), 2.59 (1H, m, H-23b), 2.60 (1H, d, *J* = 15.6 Hz, H-2′a), 2.96 (1H, d, *J* = 15.6 Hz, H-2′b), 4.29 (1H, d, *J* = 13.2 Hz, H-18a), 4.44 (1H, d, *J* = 13.2 Hz, H-18b), 4.62 (1H, dd, *J* = 10.8, 8.0 Hz, H-16), 4.71 (1H, d, *J* = 8.0 Hz, H-15), 4.85 (1H, dd, *J* = 11.4, 4.2 Hz, H-3); ^13^C NMR (pyridine-*d*_5_, 150 MHz) δ_C_ 14.0 (C-30), 15.9 (C-29), 20.7 (C-9), 21.2 (C-6), 25.3 (C-21), 25.4 (C-26), 25.9 (C-27), 26.3 (C-28), 26.6 (3′-2×CH_3_), 26.8 (C-7), 27.0 (C-11), 27.6 (C-2, 10), 29.1 (C-23), 30.4 (C-12), 30.7 (C-19), 32.2 (C-1), 40.0 (C-4, 22), 41.2 (C-3′), 45.6 (C-2′), 47.7 (C-5), 48.6 (C-8), 51.4 (C-13), 52.2 (C-14), 52.4 (C-17), 65.4 (C-18), 72.2 (C-25), 80.9 (C-3), 83.8 (C-20), 84.0 (C-16), 84.1 (C-15), 112.1 (C-24), 171.9 (C-1′), 179.7 (C-4′); HRESIMS *m/z* 655.3822 [M + Na]^+^ (Cacld. for C_36_H_56_O_9_Na *m/z* 655.3822).

#### 3.3.2. (20*S*,24*S*)-15*β*,16*β*-Diacetoxy-18,24;20,24-diepoxy-9,19-cyclolanostane-3*β*,25-diol-3-*O*-4′-(benzyloxy) carbonyl-3′,3′-dimethylsuccinate (**3b**)

Colorless needles, 45.2% yield; m.p. 188–190 °C (MeOH); [α${]}_{D}^{20}$ –4.0 (*c* 0.1, MeOH); ^1^H NMR (600 MHz, pyridine-*d*_5_) *δ*_H_ 0.18 (1H, d, *J* = 4.2 Hz, H-19), 0.50 (1H, d, *J* = 4.2 Hz, H-19), 0.57 (1H, q, *J* = 12.0 Hz, H-6a), 0.93 (3H, s, H-28), 0.95 (3H, s, H-29), 1.11 (1H, m, H-7a), 1.12 (1H, m, H-1a), 1.16 (1H, m, H-11a), 1.19 (3H, s, H-30), 1.23 (1H, m, H-5), 1.29 (3H, s, H-21), 1.31 (1H, m, H-6b, 7b), 1.40 (6H, s, 2×CH_3_-3′), 1.47 (1H, m, H-1b), 1.54 (1H, m, H-12a), 1.57 (3H, s, H-27), 1.59 (1H, m, H-8), 1.65 (1H, m, H-2a), 1.68 (3H, s, H-26), 1.85 (1H, m, H-2b), 1.96 (1H, m, H-11b), 1.97 (1H, m, H-22a), 2.08 (1H, m, H-23a), 2.13 (3H, s, COCH_3_), 2.15 (3H, s, COCH_3_), 2.73 (1H, d, *J* = 12.0 Hz, H-17), 2.80 (1H, m, H-23b), 2.82 (1H, d, *J* = 15.6 Hz, H-2′a), 2.86 (1H, d, *J* = 15.6 Hz, H-2′b), 2.96 (1H, m, H-12b), 2.99 (1H, m, H-22b), 4.51 (1H, d, *J* = 13.2 Hz, H-18a), 4.58 (1H, d, *J* = 13.2 Hz, H-18b), 4.82 (1H, dd, *J* = 12.0, 4.2 Hz, H-3), 5.35 (2H, s, H-1″), 5.69 (1H, d, *J* = 9.0 Hz, H-15), 5.96 (1H, dd, *J* = 12.0, 9.0 Hz, H-16), 7.34 (1H, m, H-5″), 7.41 (1H, t, *J* = 7.8 Hz, H-4″,6″), 7.50 (1H, d, *J* = 7.8 Hz, H-3″,7″); ^13^C NMR (pyridine-*d*_5_, 150 MHz) *δ*_C_ 15.8 (C-29,30), 19.5 (C-9), 20.6 (C-6), 21.6 (2×COCH_3_), 25.9 (3′-2×CH_3_), 26.0 (C-26, 27), 26.2 (C-28), 26.5 (C-7, 11), 27.5 (C-2), 27.6 (C-10), 28.2 (C-12), 31.2 (C-23), 31.6 (C-19), 32.1 (C-1), 32.7 (C-21), 38.6 (C-22), 39.9 (C-4), 41.2 (C-3′), 45.0 (C-2′), 46.0 (C-13), 47.0 (C-5), 47.3 (C-8), 51.8 (C-14), 56.4 (C-17), 66.7 (C-18), 66.9 (C-1′′), 73.0 (C-25), 75.4 (C-16), 80.9 (C-3), 82.3 (C-15), 87.1 (C-20), 114.6 (C-24), 128.6 (C-3″,7″), 128.8 (C-5′′), 129.3 (C-4″,6″), 137.4 (C-2″), 171.0 (COCH_3_), 171.3 (COCH_3_), 171.6 (C-1′), 176.9 (C-4′); HRESIMS *m/z* 829.4503 [M + Na]^+^ (Cacld. for C_47_H_66_O_11_Na *m/z* 829.4503).

#### 3.3.3. (20*S*,24*S*)-18,24; 20,24-Diepoxy-9,19-cyclolanostane-3*β,*15*β*,16β,25-tetraol-3-*O*-4′-(benzyloxy) carbonyl-3′,3′-dimethylsuccinate (**3c**)

Colorless needles, 53.5% yield; m.p. 201–203 °C (MeOH); [α${]}_{D}^{20}$ 12.0 (*c* 0.1, MeOH); ^1^H NMR (600 MHz, pyridine-*d*_5_) *δ*_H_ 0.29 (1H, d, *J* = 4.2 Hz, H-19a), 0.52 (1H, d, *J* = 4.2 Hz, H-19b), 0.70 (1H, q, *J* = 12.0 Hz, H-6a), 0.90 (3H, s, H-28), 0.94 (3H, s, H-29), 1.06 (1H, m, H-7a), 1.16 (1H, m, H-1a), 1.24 (1H, m, H-11a), 1.28 (2H, m, H-12), 1.29 (1H, m, H-5), 1.30 (1H, m, H-7b), 1.31 (3H, s, H-30), 1.38 (6H, s, 2×CH_3_-3′), 1.47 (3H, s, H-27), 1.50 (1H, m, H-6b), 1.53 (1H, m, H-1b), 1.58 (3H, s, H-21), 1.63 (1H, m, H-2a), 1.66 (3H, s, H-26), 1.72 (1H, m, H-22a), 1.81 (1H, m, H-8), 1.90 (1H, m, H-2b), 2.00 (1H, m, H-22b), 2.03 (1H, m, H-11b), 2.04 (1H, d, *J* = 10.8 Hz, H-17), 2.05 (1H, m, H-23a), 2.59 (1H, m, H-23b), 2.78 (1H, d, *J* = 16.2 Hz, H-2′a), 2.81 (1H, d, *J* = 16.2 Hz, H-2′b), 4.29 (1H, d, *J* = 12.0 Hz, H-18a), 4.45 (1H, d, *J* = 12.0 Hz, H-18b), 4.62 (1H, dd, *J* = 10.8, 8.0 Hz, H-16), 4.70 (1H, d, *J* = 8.0 Hz, H-15), 4.81 (1H, dd, *J* = 11.4, 4.2 Hz, H-3), 5.34 (2H, s, H-1″), 7.32 (1H, m, H-5″), 7.39 (1H, t, *J* = 7.2 Hz, H-4′′,6′′), 7.48 (1H, d, *J* = 7.8 Hz, H-3″,7″); ^13^C NMR (pyridine-*d*_5_, 150 MHz) *δ*_C_ 14.0 (C-30), 15.8 (C-29), 20.8 (C-9), 21.2 (C-6), 21.6 (2×COCH_3_), 25.4 (C-21, 26), 25.8 (C-27), 25.9 (3′-2×CH_3_), 26.2 (C-28), 26.5 (C-11), 26.8 (C-7), 27.5 (C-2), 27.6 (C-10), 29.1 (C-23), 30.4 (C-12), 30.7 (C-19), 32.1 (C-1), 40.0 (C-4,22), 41.2 (C-3′), 45.0 (C-2′), 47.7 (C-5), 48.6 (C-8), 51.4 (C-13), 52.2 (C-14), 52.5 (C-17), 65.5 (C-18), 66.7 (C-1″), 72.2 (C-25), 81.3 (C-3), 83.8 (C-20), 84.0 (C-16), 84.1 (C-15), 112.1 (C-24), 128.6 (C-3″,7″), 128.7 (C-5″), 129.3 (C-4″,6″), 137.4 (C-2″), 171.6 (C-1′), 176.9 (C-4′); HRESIMS *m/z* 745.4289 [M + Na]^+^ (Cacld. for C_43_H_62_O_9_Na *m/z* 745.4292).

### 3.4. Biological Experiments

#### 3.4.1. Anti-HIV Assay

In vitro activity was determined by a previously described HIV-1 infection assay [31]. Briefly, a 96-well microtiter plate was used to set up the screening assay using MT4 cells infected by HIV-1_NL4-3_ Nanoluc-sec virus in the presence of various concentrations of derivatives. The viral replication thus can be monitored by measuring the luciferase activity using the Promega Nano-Glo Luciferase Assay System.

#### 3.4.2. Cytotoxicity Assay

The cytotoxicity of the synthesized derivatives was determined using the CellTiter-Glo^®^ Luminescent Cytotoxicity Assay (Promega, Madison, WI, USA). MT-4 cells were cultured in the presence of various concentrations of the compounds for 3 days. The percentage of viable cells was measured by following the protocol provided by the manufacturer. The CC_50_ was derived using the Quest Graph™ IC_50_ Calculator (https://www.aatbio.com/tools/ic50-calculator, accessed on 7 October 2021).

### 3.5. 3D-QSAR Study

A total of 14 compounds of cycloartane triterpenoid type mother nuclear structure with anti-HIV activity ranging from 0.025 to 7.62 µM were selected as the dataset in our experiments. Small molecules were preprocessed using the LigPrep module of the Schrödinger Suite (Schrödinger 2018, LLC, New York, NY, USA). The whole dataset was randomly divided into the 80% training set and 20% test set. The 3D-QSAR model was constructed in the 3D field-based module of the Schrödinger software. The IC_50_ values were converted into the PIC_50_ values and used as the dependent variables for QSAR analyses. A Gaussian field style was employed and partial least-square (PLS) was used for the regression analysis. The van der Waals potential and the electrostatic potential were considered as separate terms.

### 3.6. Molecular Docking

Molecular docking was performed using the Molecular Operating Environment software (MOE 2019) (Chemical Computing Group, Montreal, Canada). The crystal structure of the CA_CTD_-SP1 Gag fragment (PDB ID: 5I4T) was obtained from the RCSB Protein Data Bank. The structure file was subjected to the QuickPrep procedure of MOE. The BVM structure was taken from the PubChem database. The 3D structure of DSC was generated using Chem3D software 2018. Multiple conformations of small molecules in the format of mol2 were generated as ligand libraries using Frog2.14 program (https://bioserv.rpbs.univ-paris-diderot.fr/services/Frog2/, accessed on 7 October 2021). The partial charges of all protein and ligand atoms were calculated using the implemented Amber10: EHT force field. For both ligands, all poses were located inside the cavity formed by the 6HB with different positions or orientations. Docking simulation was carried out choosing the triangle matcher for placement of the ligand in the binding site and ranked with the London dG scoring function. The best 30 poses were passed to refinement and energy minimization using the rigid receptor method and then rescored with the GBVI/WSA dG scoring function.

### 3.7. Molecular Dynamics Simulations

MDs were performed using the AMBER package version 22 within the Ubuntu 20.04 Linux operating system which was installed on high-performance computing cluster. All calculations were implemented using AMBER′s PMEMD engine running on CUDA-enabled GPUs. The CA_CTD_-SP1 protein adopts the ff14SB force field with the TIP3P water model and small molecules select the GAFF force field [32]. The best scored binding poses calculated by molecular docking were used as input for MD. The antechamber module (implemented in AmberTools22) was utilized for the preparation of the ligands. The protein was pretreated with the leap module. The water box and the complex system were separately optimized. The energy minimizations were performed on the solvated systems for 1000 steps using a combination of steepest descent and conjugate gradient methods. After the energy of the system converges, the optimization process will automatically end. The system was then gradually heated to 300 K and equilibrated under the NVT ensemble. MD productions were performed for 50 ns timescale in two steps for each complex, and the trajectories were gathered every 1 ps. The first step was 40 ns to ensure that the conformation had stabilized, and the second step was 10 ns for energy calculation and interaction analysis. The energy and temperature data in the simulation process were generated by Perl script, and the change trend of data was visualized by Xmgrace program. The minimum value of potential energy data was found by Awk script, then the structure was extracted by CPPTRAJ to obtain the lowest potential energy structure [33]. Visualization platform Visual Molecular Dynamics (VMD, version 1.9.4a50) was used for analysis of the MDs results [34].

The binding free energies (ignoring entropy changes) of CA_CTD_-SP1 to DSC and BVM were calculated using Amber22 by the MM/PBSA and MM/GBSA methods, which are common for the calculation of affinity energy [35]. The binding free energy was decomposed into molecular mechanics terms and solvation energies, respectively. The van der Waals forces, electrostatic interactions, polar solvation energies and apolar solvation energies were computed [36,37].

Understanding the process of ligand recognition by proteins is vital for drug design. The protein residues present at the binding site form pockets that provide a conducive environment for ligand recognition. A particular residue or cluster of residues near the pocket are called hotspot residues that will result in a large change in the binding free energy when induced to mutate to alanine [38,39,40]. Change in the binding energy caused by alanine mutation of the active site can be used to identify what residues may be important for interaction. The alanine mutation scanning experiments were performed on the amino acids with the highest energy contribution in the residue decomposition results. The amino acids with an energy difference of less than –2 kcal/mol were generally considered to be hotspot amino acids [41].

## 4. Conclusions

In summary, five new beesioside I derivatives were designed, synthesized and evaluated for the anti-HIV efficiency in vitro, among which compound **3b** showed significant antiviral activity against HIV-1_NL4-3_ with an EC_50_ value of 0.28 µM. The molecular modeling including 3D-QSAR, molecular docking and MDs techniques was employed to investigate the potential of beesioside I derivatives as MIs. Based on the 3D-QSAR model using the dataset consisting of 14 cycloartane triterpenoids and analogues, the important structural factors affecting anti-HIV activity of beesioside I derivatives were revealed. Molecular docking studies are complementary to the 3D-QSAR results, showing that DSC was bound to CA-SP1 protein through electrostatic, hydrogen bond and hydrophobic interactions with stronger binding affinity than that of the standard BVM. The results of MDs were consistent with the docking simulation, revealing that the complex of the ligand and the protein was stable within 50 ns and that residue LYS359, LYS290 and LEU363 played a major role in binding of CA-SP1 to DSC. All the results showed that DSC acted in a different way from BVM as the potential MI. Overall, the beesioside I derivative could be considered as a promising antiviral agent, worthy of further investigation.

## Data Availability

Not applicable.

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
