# Peer review of "Anti-HIV Potential of Beesioside I Derivatives as Maturation Inhibitors: Synthesis, 3D-QSAR, Molecular Docking and Molecular Dynamics Simulations"

_ijms, 2023, doi:10.3390/ijms24021430_

Round 1
Reviewer 1 Report
Dear Chinho Chen et al.,
Regarding the manuscript titled (Anti-HIV Potential of Beesioside I Derivatives as Maturation Inhibitors: Synthesis, 3D-QSAR, Molecular Docking and Molecular Dynamics Simulations), it is an interesting article, however, the following comments should be considered before further processing:
1) Line 64, change (Molecular modelling in silico) to (In silico molecular modelling).
2) Line 69, you said (molecular dynamics (MD)), then you repeated molecular dynamics many times. Please use the abbreviation all over the manuscript.
3) Line 84, you said (However, the binding modes of the promising derivative remains unknown.). Do you mean the actual X-ray structure?? Or the computational binding mode?? Please clarify and explain??
4) Line 124, you said (The 3D-QSAR models were constructed and vali- 124 dated using the dataset composed of 14 cycloartane triterpenoids and analogues with their 125 corresponding activity data reported from our laboratory). Herein, 14 compounds is a small number to build a 3D-QSAR model?! Please clarify and explain??
5) Line 163, you said (DSC showed a 163 better docking score of –8.096 kcal/mol as compared with the standard BVM (docking 164 score = –6.574 kcal/mol),). Please write the RMSD values to confirm the validity of the docking process?? As the large compounds mostly have large RMSD values!!
6) A validation process is required to confirm the validity of the docking software??
7) Line 194, you said (Overall, the RMSD value fluctuated 194 less within 50 ns….). This sentence is incorrect, 50 ns is a time and NOT a range for fluctuation!! Please revise and rephrase??
8) Transfer Figures 5, 6, and 7 to the supplementary data.
9) Line 257, you said (croton anhydride). Do you mean crotonic anhydride??!
Author Response
1) Line 64, change (Molecular modelling in silico) to (In silico molecular modelling)
Response: Thank you for your comment. The revision has been made accordingly.
2) Line 69, you said (molecular dynamics (MD)), then you repeated molecular dynamics many times. Please use the abbreviation all over the manuscript.
Response: Thank you for your comment. The revision has been made accordingly.
3) Line 84, you said (However, the binding modes of the promising derivative remains unknown.). Do you mean the actual X-ray structure?? Or the computational binding mode?? Please clarify and explain??
Response: Thank you for your comment. Because both DSC and BVM are triterpenoid derivatives and could possess similar mechanism of action, our aim is to identify plausible binding mode of DSC to potential target of maturation inhibitor like CA-SP1 by using computational approaches.
4) Line 124, you said (The 3D-QSAR models were constructed and vali- 124 dated using the dataset composed of 14 cycloartane triterpenoids and analogues with their 125 corresponding activity data reported from our laboratory). Herein, 14 compounds is a small number to build a 3D-QSAR model?! Please clarify and explain??
Response: Thank you for your comment. All the compounds used in the 3D-QSAR model were obtained via isolation or modification in our lab. To ensure the reliability of the model, all cycloartane triterpenoids in the dataset possessed similar structures and showed a certain gradient in anti-HIV activity. Only 14 molecules were eligible for the construction of the model. Generally, the number of theoretically allowed molecules in the training set are at least 10, which can be seen in the following references.
- Ying Chen, Xiaoyu Cai, Long Jiang, Yu Li. Prediction of octanol-air partition coefficients for polychlorinated biphenyls (PCBs) using 3D-QSAR models. Ecotoxicology and Environmental Safety 124 (2016) 202-212, http://dx.doi.org/10.1016/j.ecoenv.2015.10.024.
- Adib Ghale, Adnane Aouidate, Mounir Ghamali, Abdelouahid Sbai, Mohammed Bouachrine, Tahar Lakhlifi. 3D-QSAR modeling and molecular docking studies on a series of 2,5 disubstituted 1,3,4-oxadiazoles. Journal of Molecular Structure 1145 (2017) 278-284, http://dx.doi.org/10.1016/j.molstruc.2017.05.065.
5) Line 163, you said (DSC showed a 163 better docking score of –8.096 kcal/mol as compared with the standard BVM (docking 164 score = –6.574 kcal/mol),). Please write the RMSD values to confirm the validity of the docking process?? As the large compounds mostly have large RMSD values!!
Response: Thank you for your comment. DSC showed a docking score of –8.096 kcal/mol with a RMSD value of 2.73 Å, while the standard BVM had a docking score of –6.574 kcal/mol with a RMSD value of 1.56 Å.
6) A validation process is required to confirm the validity of the docking software??
Response: Thank you for your comment. Because CA-SP1 protein (PBD: 5I4T) is lack of co-crystallized ligand, the SiteFinder module in MOE was used to identify the active site, which is located in the cavity formed by the six-helix bundle, consistent with that reported in the literature [1]. Secondly, using BVM as the positive control, the docking results showed that the interaction relationship was also consistent with that reported in the literature, that is, the binding of BVM and CA-SP1 was mainly mediated by electrostatic and hydrophobic interaction, which showed that the carboxyl group at C-3 side chain of BVM formed two ion bonds with two amino acids (LYS290 and LYS359), and contacted LEU363 and MET367 through hydrophobic interaction [2,3]. In addition, the MD simulation studies can also prove the effectiveness of the docking process.
- Wagner, J.M.; Zadrozny, K.K.; Chrustowicz, J.; Purdy, M.D.; Yeager, M.; Ganser-Pornillos, B.K.; Pornillos, O. Crystal structure of an HIV assembly and maturation switch. Elife 2016, 5, e17063, https://doi.org/10.7554/elife.17063.
- Mingzhang Wang, Caitlin M. Quinn, Juan R. Perilla, Huilan Zhang, Randall Shirra Jr., Guangjin Hou, In-Ja Byeon, Christopher L. Suiter, Sherimay Ablan, Emiko Urano, Theodore J. Nitz, Christopher Aiken, Eric O. Freed, Peijun Zhang, Klaus Schulten, Angela M. Gronenborn, Tatyana Polenova. Quenching protein dynamics interferes with HIV capsid maturation. Nature Communications 2017, 8, 1779, https://doi.org/10.1038/s41467-017-01856-y.
- Purdy, M.D.; Shi, D.; Chrustowicz, J.; Hattne, J.; Gonen, T.; Yeager, M. MicroED structures of HIV-1 Gag CTD-SP1 reveal binding interactions with the maturation inhibitor bevirimat. Proc. Natl. Acad. Sci. 2018, 115, 13258−13263. https://doi.org/10.1073/pnas. 1806806115.
7) Line 194, you said (Overall, the RMSD value fluctuated 194 less within 50 ns….). This sentence is incorrect, 50 ns is a time and NOT a range for fluctuation!! Please revise and rephrase??
Response: Thank you for your comment. The original sentence has been revised to “the RMSD value fluctuated less during the MD simulation”.
8) Transfer Figures 5, 6, and 7 to the supplementary data.
Response: Thank you for your comment. Figures 5-7 have been transferred to the supplementary materials.
9) Line 257, you said (croton anhydride). Do you mean crotonic anhydride??!
Response: Thank you for your comment. It has been modified to "crotonic anhydride".

Reviewer 2 Report
Introduction
- Authors should mention the Rational behind the tested compounds and tested target in a figure
Results
- Use chem draw in scheme, I can see that scheme 1 seem to be faint figure
- In Table 1, authors should provide the dose-response sigmoidal curve for at least the most active compound, and how to calculate the CC50 value
- (+/-) should be replaced by (±)
- In molecular docking study, authors should prove the validation of the docking protocol, by have the superimposition of the co-crystallized ligand and calculation of the RMSD value
- What are the scientific information authors need figure 3 to explain, this figure needs more illustration and to be more informative.
- Authors should do the molecular visualization tool for the tested proteins to highlight the key amnio acids for interaction.
- I can see also figure 7, needs more illustration and needs to be more informative. I couldn’t see the intended information.
Author Response
- Introduction: Authors should mention the Rational behind the tested compounds and tested target in a figure
Response: Thank you for your comment. A figure (Figure 1) to explain the rationale behind the tested compounds and tested target has been added.
Results:
- Use chem draw in scheme, I can see that scheme 1 seem to be faint figure
Response: Thank you for your comment. Scheme 1 has been redrawn using ChemDraw.
- In Table 1, authors should provide the dose-response sigmoidal curve for at least the most active compound, and how to calculate the CC50 value
Response: Thank you for your comment. The following figure represents a typical dose-response curve from one test of compound 3b. The anti-HIV-1 activity of the compound was assessed with a 5-fold serial dilution of the compound encompassing the inhibitory phase of the dose-response curve. The CC50 was derived using the Quest Graph™ IC50 Calculator (https://www.aatbio.com/tools/ic50-calculator).”
- (+/-) should be replaced by (±)
Response: Thank you for your suggestions. The revision has been made accordingly.
- In molecular docking study, authors should prove the validation of the docking protocol, by have the superimposition of the co-crystallized ligand and calculation of the RMSD value
Response: Thank you for your comment. As you said, docking protocol can be validated by superimposition of re-docked co-crystalized ligands of the target. However, CA-SP1 protein is lack of co-crystallized ligands, we can’t re-dock native co-crystallized ligands into their original binding cavities and then evaluate their RMSD values. In our studies, the SiteFinder module in MOE was used to identify the active site, which was located in the cavity formed by the CA-SP1 six-helix bundle, consistent with that reported in the literature [1]. Secondly, using BVM as the positive control, the docking results showed that the interaction relationship was also consistent with that reported in the literatures [2,3]. In addition, the MD simulation studies can also prove the effectiveness of the docking process.
- Wagner, J.M.; Zadrozny, K.K.; Chrustowicz, J.; Purdy, M.D.; Yeager, M.; Ganser-Pornillos, B.K.; Pornillos, O. Crystal structure of an HIV assembly and maturation switch. Elife 2016, 5, e17063, https://doi.org/10.7554/elife.17063.
- Mingzhang Wang, Caitlin M. Quinn, Juan R. Perilla, Huilan Zhang, Randall Shirra Jr., Guangjin Hou, In-Ja Byeon, Christopher L. Suiter, Sherimay Ablan, Emiko Urano, Theodore J. Nitz, Christopher Aiken, Eric O. Freed, Peijun Zhang, Klaus Schulten, Angela M. Gronenborn, Tatyana Polenova. Quenching protein dynamics interferes with HIV capsid maturation. Nature Communications 2017, 8, 1779, https://doi.org/10.1038/s41467-017-01856-y.
- Purdy, M.D.; Shi, D.; Chrustowicz, J.; Hattne, J.; Gonen, T.; Yeager, M. MicroED structures of HIV-1 Gag CTD-SP1 reveal binding interactions with the maturation inhibitor bevirimat. Proc. Natl. Acad. Sci. 2018, 115, 13258−13263. https://doi.org/10.1073/pnas. 1806806115.
- What are the scientific information authors need figure 3 to explain, this figure needs more illustration and to be more informative.
Response: Thank you for your comment. Figure 3 has been revised into figure 4 with more illustrations and explained with detailed information. Figure 4. Docking results of CA-SP1 protein (PDB code: 5I4T) binding with ligands (green), BVM (a and c) and DSC (b and d) from a side and top view, visualized with MOE program. a, b, Side view (a, b) showing that BVM was vertically bound to the center of the 6HB in a linear extended con-formation while DSC was slightly inclined to the center of the 6HB. c, d, Top view (c, d) displaying both BVM and DSC located in the cavity formed by the CA-SP1.
- Authors should do the molecular visualization tool for the tested proteins to highlight the key amnio acids for interaction.
Response: Thank you for your comment. Molecular visualization tool VMD program has been used to create graphic as followed representing the key amino acids for interactions between CA-SP1 and DSC.
- I can see also figure 7, needs more illustration and needs to be more informative. I couldn’t see the intended information.
Response: Thank you for your comment. Figure 7 (changed to Figure S3) has been transferred to supplementary material and more illustration and information have been added. Figure S3. Lowest potential energy structure of BVM (a) and DSC (b). The graph represented the conformation with the lowest potential energy during the MDs process, which was equivalent to the most stable conformation of the molecule and had the highest probability of occurrence in the system.

Round 2
Reviewer 1 Report
Dear Respected Editor,
The authors carried out the requested modifications and responded to the comments accordingly. The paper could be accepted for publication. Best wishes.
Reviewer 2 Report
Authors have addressed all comments and the manuscript has been improved properly